# One-Year Visual Outcomes and Corneal Higher-Order Aberration Assessment of Small-Incision Lenticule Extraction for the Treatment of Myopia and Myopic Astigmatism

**DOI:** 10.3390/jcm11216294

**Published:** 2022-10-26

**Authors:** Carter J. Payne, Courtney R. Webster, Majid Moshirfar, Jaiden J. Handlon, Yasmyne C. Ronquillo, Phillip C. Hoopes

**Affiliations:** 1Case Western Reserve University School of Medicine, Cleveland, OH 44106, USA; 2Hoopes Vision Research Center, Hoopes Vision, 11820 S. State St., Ste. 200, Draper, UT 84020, USA; 3Michigan State University College of Osteopathic Medicine, East Lansing, MI 48824, USA; 4John A. Moran Eye Center, Department of Ophthalmology and Visual Sciences, University of Utah School of Medicine, Salt Lake City, UT 84132, USA; 5Utah Lions Eye Bank, Murray, UT 84107, USA

**Keywords:** SMILE, myopia, astigmatism, refractive surgery, wavefront aberrometrymyopic astigmatism

## Abstract

We present a retrospective, single-center report of one-year visual outcomes for Small Incision Lenticule Extraction (SMILE) to treat myopia and myopic astigmatism, as well as to compare outcomes with other published literature, including results from the United States Food and Drug Administration (US FDA). A total of 405 eyes with a mean preoperative spherical equivalent of −5.54 diopters (D) underwent SMILE between April 2017 and April 2022. The outcomes measured included visual acuity, manifest refraction, vector analysis, and wavefront aberrometry at various time points, specifically pre-operative and twelve months post-operatively. Results were compared to other similar published studies of SMILE outcomes between 2012 and 2021. A total of 308 and 213 eyes were evaluated at three and twelve months, respectively. At twelve months, 79% of eyes achieved UDVA ≥ 20/20, and 99% had ≥20/40, with no patients losing ≥2 lines of vision. For accuracy, 84% of eyes were within 0.5 D of target SEQ, and 97% were within 1 D. Total corneal higher order aberrations (HOA) increased from 0.33 to 0.61 um. Significant change was found in vertical coma and spherical aberration at twelve months. SMILE remains a safe and effective treatment for myopia and myopic astigmatism. Clinical outcomes are likely to improve with increased surgeon experience and refinement of technology and nomograms.

## 1. Introduction

Small-incision lenticule extraction (SMILE) is a form of laser refractive surgery which uses a femtosecond laser to carve out a small lens-shaped piece of intrastromal corneal material that is manually removed through a small incision. This procedure does not require flap formation, removal of epithelium, or excimer laser ablation, distinguishing it from other techniques such as LASIK and PRK [1]. SMILE has some key potential advantages over LASIK, including maintenance of biomechanical stability of anterior cornea, lack of flap-associated complications, and better protection or less disruption of cornea nerve fibers [2]. Though SMILE has been available internationally for over a decade, only relatively recently has it been approved in the United States [3]. SMILE using the VisuMax (Carl Zeiss Meditec, Jena, Germany) femtosecond laser was FDA approved for the treatment of simple myopia in 2016, then for treatment of myopic astigmatism in 2018 [4]. Various studies have compared SMILE to LASIK, finding comparable results in terms of efficacy, safety, and stability [5]. Studies of SMILE visual outcomes in the United States since the FDA clinical trial are lacking, particularly studies of larger cohorts with one-year follow-up. As such, we report a single-site experience using SMILE and compare the results with the FDA data and existing literature.

## 2. Materials and Methods

*Study Design*: This was a non-randomized retrospective cohort study performed by chart review of de-identified medical records at a single site refractive surgery center in Draper, Utah. Inclusion criteria were patients with a diagnosis of myopia and myopic astigmatism who underwent SMILE at our center between April 2017 and April 2022. A total of 405 eyes of 207 patients were treated with SMILE. Collected data includes date of procedure, baseline preoperative distance visual acuity both corrected and uncorrected (BCVA and UDVA, respectively), preoperative manifest refraction, and VisuMax laser settings. Primary outcome measures include post-operative BCVA, UDVA, and manifest refraction values. Secondary outcome measures include post-operative complications and change in corneal wavefront aberrometry values. Post-operative visual acuity (BCVA/UDVA) and manifest refraction values were collected where available for follow-up visits at 1 day, 1 week, 1 month, 3 months, 6 months, and 12 months post-operative, as well as any visits beyond 1 year. Visual acuity measurements were performed using electronic Snellen letter charts, and if both eyes of the same patient were to be tested in the same visit, the letters were randomized between measurements to ensure no memorization could occur. There were 22 monovision eyes that were excluded from post-operative UDVA analysis but were included in all other analysis. Wavefront aberrometry values were collected for 41 eyes using a rotating Scheimpflug camera (Pentacam HR; Oculus, Wetzlar, Germany), which measured total corneal root mean square (RMS) values of higher order aberrations (HOAs) over a 6 mm central diameter, including spherical aberration, horizontal and vertical coma, and horizontal and oblique trefoil.

*Surgical Procedure*: SMILE was performed at a single site with the VisuMax 500 kHz femtosecond laser (Carl Zeiss Meditec, Jena, Germany). Preoperative limbal markings were placed with a surgical marking pen at 3 and 9 o’clock on patients seated in an upright position, after which the patient was taken to the operating room and placed in a supine position. The cornea was marked again inside the limbus at the corresponding markings at 3 and 9 o’clock with a caliper set at 8 mm. Careful observation was taken to note any cyclotorsion after application of the vacuum. Manual adjustment was performed if required with either clockwise or counterclockwise rotation to align the respective markings with the horizontal lenticule on the laser scope. The cap thickness was set at 120 μm and a cap diameter of 7.5 mm. Hinge location was superior with the hinge angle set to 60 degrees and side-cut angle of 90 degrees. Lenticulate diameter was either 6.0 or 6.5 mm depending on surgeon discretion previously calculated for each patient. Spot separation was 3.70 μm for the lenticule, 2.00 μm for the lenticule side-cut, 3.80 μm for the cap, and 2.00 μm for the cap side. The laser-bed energy was set at 125 nJ. Postoperatively, patients were prescribed ofloxacin 0.3% or moxifloxacin 0.5% four times a day for one week along with prednisolone acetate 1.0% four times daily for one week with a weekly taper. 

*Statistical Analysis*: Patient data were collected and entered into Datagraph-Med (Wendelstein, Germany) for database creation only, then exported to EXCEL (Microsoft Corporation, Redmond, WA, USA) for basic statistical analysis and graph creation. Visual acuity data were collected as Snellen lines and converted to logarithm of the minimum angle of resolution (LogMAR) and decimal values for analysis where noted. Visual acuity of counting fingers (CF) was converted to 2.0 LogMAR [6]. SPSS software version 29.0 (IBM Inc., Armonk, NY, USA) was used for statistical analysis. Shapiro–Wilk test was used to assess normality. Comparison of pre- and post-operative data were made using paired and unpaired *t*-tests for means of normally distributed data, and the Mann–Whitney U test was used to compare non-normally distributed data. *p*-values less than 0.05 were considered statistically significant. To minimize bias from inclusion of bilateral eyes in HOA analysis, a random subset of unilateral eyes was selected and results were compared to the contralateral subset. Only differences that were significant for both subsets were reported as significant in the bilateral analysis. Vector analysis was performed according to the Alpins method and terminology [7], and double-angle vector diagrams were created using ASSORT^®^ Group Analysis Calculator, with manifest refraction values converted to the corneal plane using a back vertex distance of 12 mm. Astigmatic vector analysis included only patients treated for astigmatism, i.e., patients with >−0.5 D manifest cylinder who received no laser treatment for astigmatism were excluded. Definitions for terms in vector analysis include the following: target induced astigmatism (TIA) is the intended astigmatic change, while surgically induced astigmatism (SIA) is the actual astigmatic change after surgery; the difference vector (DV) is the difference between the SIA and TIA vectors and represents residual astigmatism, while magnitude of error (ME) is their arithmetic difference; the angle between SIA and TIA is called the angle of error; the ratio of SIA to TIA is known as the corrective index (CI) and should be equal to one in an ideal outcome; CI × 100% gives the percent of corrected astigmatism [8].

## 3. Results

### 3.1. Demographics

SMILE was performed on 405 eyes of 207 patients, with 41% male and 59% female, and 50.1% right eyes and 49.9% left eyes. As of the date of writing of this manuscript, 128 eyes do not yet have 12-month visit data. Pre-operative corrected distance visual acuity (CDVA) was 20/20 or better for all patients. Of note, 22 eyes were targeted for monovision with the remainder targeted for plano. PRK enhancement after SMLE was performed in 15 eyes of 11 patients. A total of 237 eyes were treated for spherical myopia only and 168 eyes were treated for astigmatic myopia. Baseline patient demographics are displayed in Table 1.

The average laser setting for sphere was −5.45 ± 1.50 D, average cylinder laser setting was −0.54 ± 0.66 D, and 22% of patients were treated with a 6.0 mm optical zone (OZ) and 78% with 6.5 mm OZ. Manifest refraction and visual outcomes (UDVA and CDVA in LogMAR) are shown for preoperative, 3-month, 12-month, and >1-year visits in Table 2. Preoperative manifest sphere and cylinder ranged from −10.75 to −1.5 and −3.25 to 0, respectively. The 3-month and 12-month manifest refraction and visual outcomes are displayed together for comparison in the standard 9 graphs (Figure 1).

### 3.2. Efficacy, Visual Acuity, and Safety

Only eyes with a plano target were included in visual acuity reporting and 22 eyes with monovision target refractions were excluded. At twelve months, 79% of eyes had UDVA of 20/20 or better. There were 92% and 91% of eyes with UDVA of 20/25 or better at 3 and 12 months, respectively (Figure 1A). A total of 100% of eyes had a pre-operative CDVA of 20/20 or better. When comparing UDVA postoperatively, 70% were the same or better than their CDVA at 12 months and 90% were within 1 line as demonstrated in Figure 1B. At 3 months, 3% of eyes had lost 1 line or more in CDVA and this increased to 10% at 12 months. The efficacy index at twelve months was 0.94 among 211 eyes (see Table 2). There were 0.3% that had lost 2 or more lines of CDVA at 3 months with no eyes losing more than 2 lines at 12 months. At 3 and 12 months, 16% and 19% of eyes, respectively, gained 1 line of CDVA (Figure 1C). Safety index was greater than 1 at three and twelve months with 1.04 and 1.03, respectively (Table 2).

### 3.3. Accuracy and Stability

Attempted versus achieved spherical equivalent refraction can be found in Figure 1D. The percent of postoperative manifest spherical equivalent (MSE) within ±1.00 D of target at 3 and 12 months was 94% and 97% of eyes, respectively. Over the same time periods the MSE within ±0.50 D was 84% at both time points (Figure 1E). MSE was relatively stable from 3 to 12 months with 13% of eyes that had a recorded MSE at both time points having a change greater than 0.50 D (Figure 1F).

### 3.4. Astigmatism and Vector Analysis

At three and twelve months postoperatively, 97.4% and 98.1% of eyes had ≤1.00 D refractive astigmatism, respectively (Figure 1G). The mean refractive cylinder (±SD) at three and twelve months presented as −0.38 (±0.33) D and −0.40 (±0.32) D (Table 2). The mean target induced astigmatism (TIA) at three months after surgery was calculated to be 0.97 ± 0.45 D and 1.00 ± 0.46 D at twelve months. At three and twelve months postoperatively, the mean SIA was close to the TIA with 0.92 ± 0.51 D and 0.91 ± 0.50 D, respectively (Figure 1H). Double angle vector diagrams (DAVD) were created utilizing the ASSORT ^®^ Group Analysis Calculator in plus cylinder power as shown in Figure 2. The difference vector centroid magnitude between TIA and SIA at three and twelve months was 0.18 D axis 178 and 0.24 D axis 180, respectively. When compared to a preoperative mean cylinder power of −0.621 D the three-month postoperative mean cylinder power was −0.38 D. This remained stable at twelve months with a mean of −0.40 D. The refractive geometric mean for correction index (CI) at three months was 0.87, and 0.83 at 12 months (see Figure 2). This indicates a trend of undercorrection by 13–17%. This is also the trend observed in Figure 1H when comparing TIA vs. SIA as the line of best fit for both 3- and 12-month trends towards undercorrection and the slope for each are less than one, representing relative undercorrection. Refractive astigmatism angle of error at twelve months showed an overall arithmetic mean of 0.85 ± 17.46 degrees with 78.3% of eyes falling between −15° and 15° (Figure 1I).

### 3.5. Long-Term Outcomes beyond 12 Months

To date, we have 31 patients with follow-up visits beyond 12 months, ranging from 18 months to 5 years post-operatively. Average visual acuity remains generally stable for these patients with a slight worsening from earlier time points as shown in Table 2. Two eyes from one patient had a significant decline in CDVA at three years, but chart review revealed formation of significant posterior subcapsular cataracts that were subsequently removed, restoring the patient’s CDVA to >20/20 OU at 3 ½ years post-SMILE.

### 3.6. Corneal Wavefront Aberrometry

41 eyes underwent a corneal tomography scan at 12 months, including wavefront analysis with 6.0 mm scan sizes. Mean and standard deviation for pre- and post-operative root mean square (RMS) values are shown in Figure 3. The Shapiro–Wilk test showed no evidence for non-normality in each pre- and post-operative sample (*p* > 0.05). RMS values for total corneal higher-order aberrations (HOA) significantly increased pre-operatively to 12 months (0.33 to 0.61; paired *t*-test, *p* < 0.001). Vertical coma (Z3, −1) significantly decreased from −0.13 to −0.35 (*p* < 0.001), and spherical aberration significantly increased from 0.18 to 0.29 (*p* < 0.001). No significant change was found in horizontal coma or trefoil.

### 3.7. Adverse Events and Complications

One patient experienced suction loss and was converted to LASIK, thus was not included in our analysis. Two eyes experienced post-operative epithelial ingrowth. One patient experienced a small tear-out of the incision. Two eyes experienced diffuse lamellar keratitis. A total of 15 eyes of 11 patients required PRK enhancement post-SMILE.

## 4. Discussion

This retrospective study represents one of the largest cohorts of SMILE patients from a single site in the United States since FDA approval in 2016, with outcomes at one year and beyond. Our visual outcomes are generally comparable to other published studies from large-volume refractive surgery groups, as well as to studies evaluating long-term outcomes beyond 12 months, as shown in Table 3.

The United States Food and Drug Administration (FDA) has established major safety endpoints and target values for refractive surgery lasers including the following: less than 5% of subjects should lose >2 lines BCVA; less than 1% of subjects should have BCVA < 20/40; minimum 85% of subjects must achieve UCVA 20/40 or better; minimum 75% of subjects should achieve refraction within ±1.00 D of target, and 50% within ±0.50 D of target; minimum 95% of subjects should have a stable manifest refraction [9]. Our results fall well within each of these endpoints. 

In comparison to the outcomes reported in the FDA SSED for VisuMax, our demographics were more evenly split with gender and eye laterality, and our patient cohort had a slightly greater average age and age range (34.5 years ranging 18–57 years vs. 33.1 years ranging 22–59 for the FDA) [4]. Our average pre-operative spherical equivalent was similar to that of the FDA trial (−5.54 D and −5.48 D, respectively), however our patients exhibited a slightly greater range of preoperative sphere (range −10.75 D to −1.5 D) and cylinder (range −3.25 D to 0 D) compared to the FDA (sphere −10 D to −1 D; cylinder −3 D to 0 D) which may explain in part the slightly less optimal outcomes at our center due to the greater difficulty in treating higher absolute refractive errors.

The software available on VisuMax at our center has been more rigid and possibly not as customizable as it is at some international laser refractive centers. The laser settings were initially less flexible and higher-powered, with a laser-bed energy of 145 nJ in 2018, updated now to 125 nJ as the currently lowest available energy setting. The spot separation has also decreased from 4.4 μm to 3.70 μm for the lenticule. There was also no nomogram available at our center to aid in surgical planning for the first 40 eyes, and then, after FDA approval for treatment of the astigmatic component, the nomogram required updating again after we had sufficient experience with astigmatic myopic eyes. To evaluate the possibility of improvement after refinement of laser settings and nomograms, we analyzed visual outcomes for two separate cohorts of patients treated at our center: those treated in 2017–2018 (the first patients after FDA approval), and those treated more recently in 2020–2021. Pre-operative UCVA was similar between both groups (1.45 and 1.49 LogMAR, respectively; Mann–Whitney test, *p* = 0.212), while UCVA at one day post-operatively was significantly better for the latter cohort (0.211 LogMAR in 2017–2018, 0.097 LogMAR in 2020–2021; Mann–Whitney test, *p* < 0.001). However, at one year post-operatively, there was no significant difference in UCVA (0.015 LogMAR vs. 0.017 LogMAR; Mann–Whitney test, *p* = 0.486), but it must be noted that loss to follow-up was significant in the 2020–2021 cohort. The 2017–2018 group had 73 eyes pre-operatively, of which 64 (88%) were seen at a one-year visit, while in the 2020–2021 cohort, only 68 (56%) of 121 eyes were seen at their one-year visit. It is possible that a number of the patients lost to follow-up were satisfied with their visual outcomes and did not feel the need to come for a one-year visit, thus the one-year outcomes for the 2020–2021 cohort may be biased towards patients with less-desirable visual outcomes. Schallhorn et al. have suggested this possibility, that patients who are satisfied with their outcomes are less likely to be as engaged with follow-up care, including satisfaction surveys [10]. In addition, many of our patients would have had their follow-up visits scheduled during the initial height of the COVID−19 pandemic, thus unnecessary clinic visits may have been canceled or neglected given the circumstances.

Despite some initial limitations to flexibility in surgical planning and the learning curve associated with a new procedure, our patient outcomes remain satisfactory and comparable to other reported studies. When we compare our three-month and 12-month outcomes (see Figure 1), stability remains excellent with no sign of myopic regression, and similar trends in terms of safety and efficacy hold up between follow-up time points. Comparison of our three-month follow-up cohort to other studies reporting the same timeframe shows variability in preoperative refractive values (SEQ −4.33 D to −7.19 D) with ours near the middle of the range. Certain outcomes are not reported uniformly across studies, but we report a smaller percentage of patients with better than 20/20 vision at three months relative to other studies, but equal or higher percentage of patients with better than 20/40 vision, while we also report a very small percentage of patients with loss of 2 or more lines at three months, even less than most other reported studies. Comparison of our 12-month follow-up group to the literature shows similar preoperative SEQ across studies, with our patients having lower percentage better than 20/20 but very similar percentage better than 20/40 vision. At 12 months, our SEQ accuracy is slightly lower than other reported studies with similar numbers of eyes at 84% within 0.5 D and 97% within 1 D of target refraction. For astigmatic outcomes, our trend of undercorrection is generally consistent with other studies. A previously published review of the literature found variability in correction index after SMILE, with the majority ranging from 0.84 to 0.97 (ours being 0.87 and 0.83 at three and 12 months), and a general undercorrection of around 10–15% [8]. These other studies, mostly international, have possibly had longer experience with SMILE, and have thus refined their technique and preoperative planning with laser settings. We hope to attain better outcomes with more experience, particularly regarding percent UDVA ≥ 20/20 and SEQ accuracy. 

Regarding wavefront aberrometry, on average, our patients had a mean increase in total corneal HOA RMS after one year (see Figure 3). The increase from 0.33 µm to 0.61 (mean change 0.28 µm ± 0.17) after one year was similar to Jin et al. who reported a change from 0.39 to 0.66 in a low myopia group, and 0.40 to 0.99 in a high myopia group after three months [11]. Sekundo et al., on the other hand, reported a smaller increase from 0.17 ± 0.08 to 0.27 ± 0.1 µm at one year, however they measured in a 5.0 mm zone rather than 6.0 mm in our data [12]. Data from the FDA SSED also shows an even less dramatic increase in HOA after one year, with a reported mean increase in HOA RMS of only 0.088 ± 0.336, though only mean change in RMS was reported, not mean pre- and post-operative values, limiting the extent of our comparison. For specific aberration changes, we report a significant increase in magnitude of vertical coma, but no significant change in horizontal coma. A change in vertical coma but not horizontal may suggest a tendency for decentration in the vertical axis but not in the horizontal, as supported by Li et al. [13], though results from other studies are conflicting as to the strength of the association between direction of decentration and type of coma [14]. There was also a statistically significant increase in spherical aberration. It should be noted, however, that only 20% of our patients with one-year follow-up had Pentacam tomography performed, potentially introducing bias in these types of outcomes.

Outcomes at visits greater than 1 year have been reported by various centers, with follow-up data at various time points, including 2, 3, and 5 years. Due to the relatively young nature of the SMILE procedure in the US, long-term data is lacking in US centers. This limits our ability to identify risk for myopic regression or corneal ectasias after SMILE, and continuous long-term follow-up over 5+ years is warranted for these patients. Our patient outcomes show a slight decline after >1 year compared to earlier follow-up points, with a small reduction in both efficacy and safety indices. We can report >1-year outcomes for only 30 eyes with a range of follow-up periods from 1.5 to 5 years post-operatively, leaving the data more vulnerable to possible outliers and follow-up bias similar to reported above. As time goes on and more patients who received SMILE after our refined nomogram and increased experience, we expect these long-term outcomes to improve. There is significant variability in the various outcomes between different studies (shown in Table 3), for example safety measures (20/20 UDVA or better) have a range of 63% to >90%, and spherical equivalent accuracy (within 0.5 D SEQ) ranges from 48% to 92% depending on the study. All reported studies have relatively small numbers of eyes (range 31–87 eyes). As experience with SMILE increases globally and nationally, patterns in long-term outcomes should become clearer.

**Table 3 jcm-11-06294-t003:** Comparison of current study with SMILE outcomes reported in the literature.

Study	Year	Preop SEQ (SD)	Preop Sph (SD)	Preop Cyl (SD)	n	%UDVA ≥ 20/20	%UDVA ≥ 20/40	%CDVA Loss ≥ 2 lines	%SEQ ≤ 0.50 D	%SEQ ≤ 1.00 D
3 months										
Current Study	2022	−5.59 (1.62)	−5.29 (1.60)	−0.62 (0.60)	308	80%	99%	0.30%	84%	94%
Ivarsen and Hjortdal [15]	2014	---	---	<−2.50 D	669	---	*81% ≥20/25*	2.80%	---	---
				≥−2.50 D	106	---	*64% ≥20/25*	0.90%	---	---
Hjordtal et al. [16]	2012	−7.19 (1.30)	−6.41 (1.77)	−0.60 (0.46)	670	---	---	---	80.10%	94.20%
Hansen et al. [17]	2016	−6.82 (1.66)	---	−0.83 (0.84)	722	---	*83% ≥20/25*	1.60%	88%	98%
Kamiya et al. (multi- center) [18]	2019	−4.33 (1.61)	---	−0.64 (0.51)	252	100%	100%	0.80%	88%	98%
Liu et al. (3–6 mo) [19]	2021	−4.96 (2.07)	---	−1.04 (0.77)	462	93%	---	0%	98%	100%
12 months										
Current Study	2022	−5.55 (1.48)	−5.22 (1.43)	−0.58 (0.61)	213	79%	99%	0%	84%	97%
VisuMax FDA PMA [4]	2018	*−5.48*	−4.82 (2.39)	−1.34 (0.80)	349	89.40%	98.90%	0%	94.80%	99.10%
Dishler et al. [20]	2020	−5.41 (2.31)	−4.65 (2.33)	−1.52 (0.70)	300	89.00%	99.00%	0%	95.30%	99.30%
Chansue et al. [21]	2015	−4.96 (1.88)	−4.61 (1.85)	−0.71 (0.61)	318/347	90%	98%	0%	93%	99%
Wu et al. [22]	2016	<−6.0 D	−6.49 (0.93)	−0.82 (0.68)	65	---	---	---	95.40%	100%
		≥−6.0 D	−4.21 (1.10)	−0.95 (0.88)	91	---	---	---	96.70%	100%
>1 year										
Current Study (1.5–5 yrs)	2022	−4.91 (1.46)	−4.73 (1.45)	−0.36 (0.42)	31	63%	96.70%	0%	77.80%	96.30%
Han et al. (3 yrs) [23]	2019	−6.54 (1.69)	−6.14 (1.62)	−0.80 (0.68)	60	90%	---	*2% lost 1 line*	80%	---
Yildirim et al. (2 yrs) [24]	2016	−7.10 (0.95)	−6.64 (0.88)	−0.82 (0.55)	45	80%	100%	*2% lost 1 line*	92%	100%
Tian et al. (5 yrs) [25]	2022	−7.16 (1.51)	−6.74 (1.45)	−0.84 (0.70)	41	90.20%	100%	0%	87.80%	95.10%
Pedersen et al. (3 yrs) [26]	2015	−7.30 (1.40)	---	−0.70 (0.60)	87	72%	---	---	78%	90%
Blum et al. (5 yrs) [27]	2016	---	---	---	56	---	---	0%	48.20%	78.60%

*Limitations*: A major limitation of this study, as referred to earlier, is loss to follow-up for a significant number of patients at longer-term visits (12 months and beyond), a common issue for retrospective studies. Reduced patient numbers at later visits results in a smaller sample size and can potentially introduce bias. Additionally, we must consider that a certain proportion of patients have not yet reached their 12-month mark with many receiving SMILE within just a few months of the writing of this manuscript. A total of 128 eyes had not yet reached one year since their SMILE at the time of our data collection (cut-off August 2021). Other limitations inherent to the retrospective nature of our study include missing data points for some proportion of patients for various reasons, as well as the lack of a control group. Another limitation is the potential for bias from inter-eye correlation with the inclusion of bilateral eyes from most patients, however, we felt it necessary to include all eyes to maintain an adequate sample size and report total outcomes. Despite the limitations listed, we are still able to report good visual outcomes with SMILE at our site after one year for patients with myopia and myopic astigmatism.

## 5. Conclusions

In this study, we have presented patient visual and refractive outcomes after SMILE for the first 405 eyes treated at our surgery center. We report satisfactory results that continue to support SMILE as being a safe and effective treatment for myopia and myopic astigmatism. We encourage other refractive surgery centers to continue reporting SMILE outcomes, especially over the long-term, to develop a more robust understanding of the effects and success of SMILE.

## Figures and Tables

**Figure 1 jcm-11-06294-f001:**
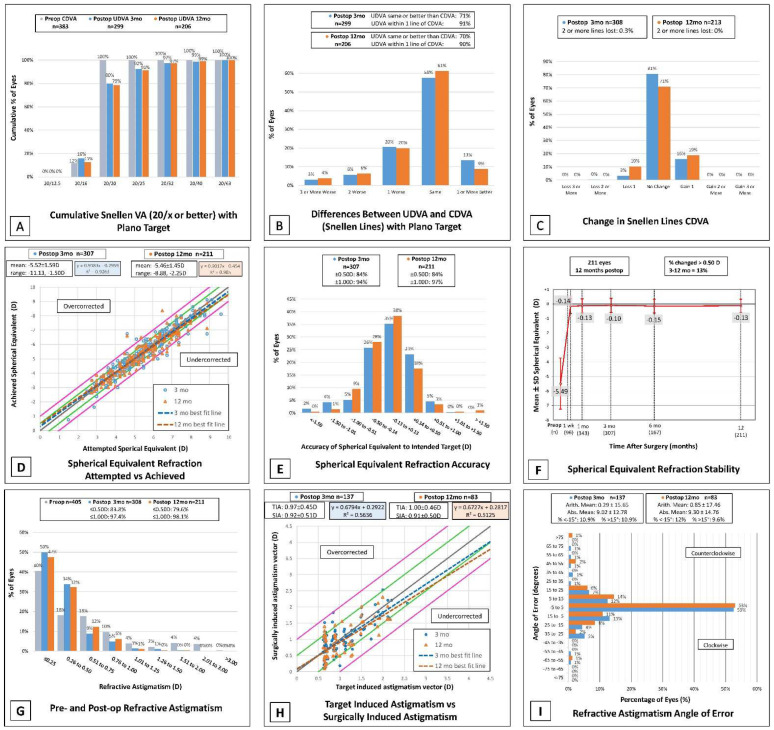
Nine standard graphs for reporting corneal refractive surgery outcomes at three and 12 months post-operatively. (**A**) Post-operative UDVA versus pre-operative CDVA. (**B**) Change in Snellen lines from pre-operative CDVA to post-operative UDVA. (**C**) Change in Snellen lines from pre-operative CDVA to post-operative CDVA. (**D**) Attempted versus achieved spherical equivalent refraction, with linear regression and correlation values; the grey line represents the equation y = x, and the closer the regression line is to the grey line, the more accurate the results; the green lines mark ±0.5D and the pink lines mark ±1D of the grey line. (**E**) Accuracy of post-operative spherical equivalent refraction to target. (**F**) Stability of spherical equivalent refraction, shown as the trend of mean SEQ at preop, one week, one month, three months, six months, and one year post-operatively. (**G**) Change in refractive astigmatism. (**H**) Target-induced astigmatism (TIA) vs. surgically induced astigmatism (SIA); see graph D for explanation of colored lines. (**I**) Histogram of refractive astigmatism angle of error.

**Figure 2 jcm-11-06294-f002:**
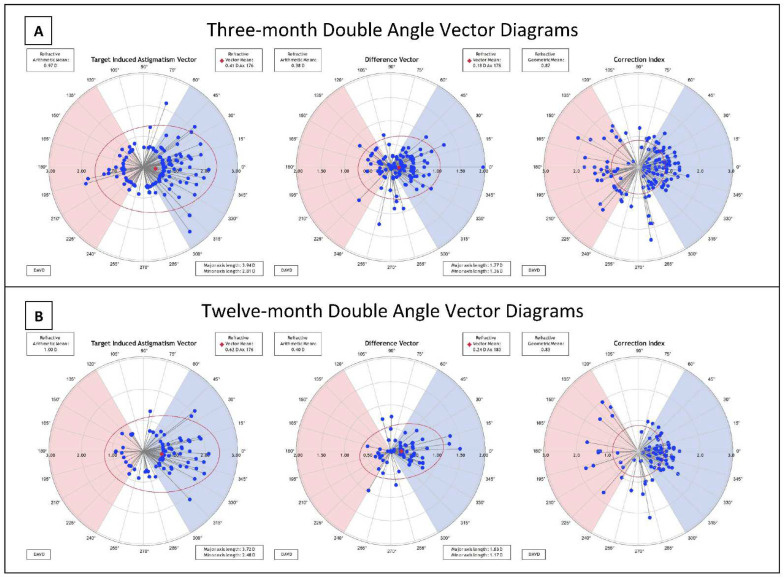
Double-angle vector diagrams for target-induced astigmatism, difference vector, and correction index at three (**A**) and 12 (**B**) months post-operatively. *Note*: Angles are doubled. Vectors are shown in plus cylinder power. TIA vectors are perpendicular to patient astigmatism. Vectors in blue shaded areas for TIA show treatment of with-the-rule astigmatism. Refer to results section for explanation of terms (TIA, DV, CI).

**Figure 3 jcm-11-06294-f003:**
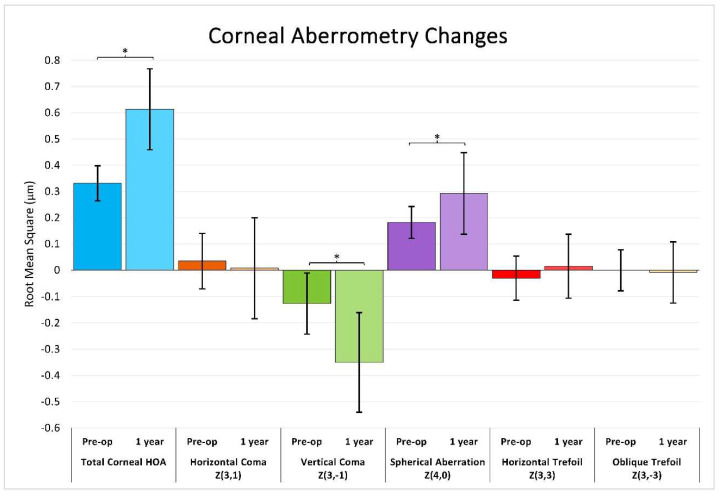
Pre- and one-year post-operative RMS values for select higher-order aberrations. Error bars represent standard deviation. Paired *t*-tests were used to determine significant difference between pre- and post-operative means. * Signifies a statistically significant difference (*p* < 0.05) between pre- and post-operative values.

**Table 1 jcm-11-06294-t001:** Baseline Patient Demographic and Refractive Data.

Demographics	Treated for Spherical Myopia Only	Treated for Astigmatic Myopia	All Treated Eyes	Targeted for Monovision	Required PRK Enhancement
**Number of eyes**	217 eyes	188 eyes	405 eyes	22 eyes	15 eyes
(no. of patients)	(129 patients)	(116 patients)	(207 patients)	−5.40%	(11 patients)
**Gender**					
Male	52 (40%)	43 (37%)	85 (41%)	9 (41%)	3 (20%)
Female	77 (60%)	73 (63%)	122 (59%)	13 (59%)	8 (80%)
**Surgical Eye**					
OD	111 (51%)	92 (49%)	203 (50%)	4 (18%)	6 (40%)
OS	106 (49%)	96 (51%)	202 (50%)	18 (82%)	9 (60%)
**Age (years)**					
Mean (SD)	34.4 (7.6)	34.6 (7.9)	34.5 (7.7)	46.8 (4.5)	34.2 (6.2)
Min., Max.	18, 55	21, 57	18, 57	39, 54	25, 45
**MRSE**	−5.27 (1.55)	−5.84 (1.53)	−5.55 (1.57)	---	---
Average sphere	−5.16 (1.55)	−5.30 (1.51)	−5.23 (1.54)	---	---
Average cylinder	−0.22 (0.22)	−1.10 (0.62)	−0.63 (0.63)	---	---

**Abbreviations**: PRK: photorefractive keratectomy; OD: right eye; OS: left eye; SD: standard deviation; MRSE: manifest refraction spherical equivalent.

**Table 2 jcm-11-06294-t002:** Refractive and VA outcomes over time.

Outcomes	Preoperative	3 Months	12 Months	>1 Year (Range: 18 mo–3 yr)
**CDVA (LogMAR)**	n = 383	n = 309	n = 211	n = 26
Mean (SD)	−0.007 (0.027)	−0.015 (0.042)	−0.022 (0.046)	0.015 (0.087)
Range (Min, Max)	−0.14, 0.04	−0.14, 0.18	−0.14, 0.06	−0.02, 0.44
**UDVA (LogMAR)**	n = 376 *	n = 299	n = 207	n = 30
Mean (SD)	1.43 (0.24)	0.026 (0.099)	0.025 (0.091)	0.08 (0.134)
Range (Min, Max)	0.1, 1.7	−0.14, 0.8	−0.14, 0.54	0, 0.54
**Sphere**	n = 405	n = 309	n = 211	n = 27
Mean (SD)	−5.23 (1.54)	0.088 (0.486)	0.075 (0.447)	−0.120 (0.487)
Range (Min, Max)	−10.75, −1.5	−2, 1.5	−1.5, 2.25	−1.0, 0.75
**Cylinder**	n = 405	n = 309	n = 211	n = 27
Mean (SD)	−0.621 (0.606)	−0.375 (0.334)	−0.400 (0.326)	−0.417 (0.277)
Range (Min, Max)	−3.25, 0	−2, 0.5	−1.75, 0	−1, 0
≤0.50 D (%)	---	83.80%	78.60%	81.40%
≤1.00 D (%)	---	97.40%	97.10%	100%
**MSE**	n = 405	n = 309	n = 211	n = 27
Mean (SD)	−5.54 (1.57)	−0.099 (0.489)	−0.127 (0.451)	−0.329 (0.460)
Range (Min, Max)	−11.125, −1.5	−2.125, 1.25	−1.75, 1.875	−1.25, 0.625
±0.50 D of Intended (%)	---	84.10%	84.80%	77.80%
±1.00 D of Intended (%)	---	94.80%	95.20%	96.30%
**Efficacy Index ****	---	0.944 (0.182)	0.939 (0.171)	0.834 (0.174)
**Safety Index *****	---	1.037 (0.124)	1.026 (0.123)	0.979 (0.091)

Notes: * Only eyes with plano refractive targets were considered for uncorrected distance visual acuity calculations (i.e., monovision eyes were excluded). ** Efficacy index = postop UDVA/preop CDVA. *** Safety index = postop CDVA / preop CDVA. **Abbreviations:** SD: standard deviation; CDVA: corrected distance visual acuity; UDVA: uncorrected distance visual acuity; MSE: manifest spherical equivalent.

## Data Availability

Data sharing is not applicable to this article as no datasets were generated or analyzed during the current study.

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
