# Peer review of "One-Year Visual Outcomes and Corneal Higher-Order Aberration Assessment of Small-Incision Lenticule Extraction for the Treatment of Myopia and Myopic Astigmatism"

_jcm, 2022, doi:10.3390/jcm11216294_

Round 1

Reviewer 1 Report

An impressive and detailed manuscript of a new refractive surgery procedure (SMILE) that has only been FDA approved in the US for a few years.

This paper represents one of the largest cohorts of SMILE patients from a single site in the US with one year follow-up. In addition, the authors summarized the previous SMILE studies in a comprehensive table. They concluded that SMILE is a safe and effective surgical procedure to treat myopia and myopic astigmatism.

The study is precisely designed and executed. The figures in the paper are very clear and professionally edited.

Some specific questions:

-The authors mentioned few adverse events. However, it is also well known, that after SMILE, wound healing and visual recovery is slower compared to other refractive surgeries. Have the authors experienced similar results?

-How was the postoperative refraction planned/adjusted when both eyes of the patient were treated?

-Major concern of refractive surgery are myopic regression, astigmatic under correction and post-refractive ectasia. A brief comparison would be useful for everyday clinical practice, what can be expected after SMILE compared to other refractive procedures (especially LASIK)?

Reviewer 2 Report

To the authors to present the “One-Year Visual Outcomes and Corneal Higher-Order Aberration Assessment of Small-Incision Lenticule Extraction for the Treatment of Myopia and Myopic Astigmatism”

We are facing a well-worked and presented paper, but with a lot of limitations that the authors themselves describe:

#1 It is a good job to know the results of the first patients operated on in your centre, but it does not provide data that are not already reported in the literature and with much longer follow-up.

#2 It should be a prospective, not retrospective, analysis, as it biases the results.

#3 The follow-up of the patients is only one year, and not for all of them. Insufficient time to have myopia appear again if you look at studies of other types of refractive surgery.

#4 Table 3 compares more than one year of follow-up (how long is not specified), with works of 2, 3 and even 5 years. The evolution does not have to be the same, they are not comparable.

# 5 So that there is no bias in the VA results, the two eyes of the same patient should not be included, and if so, it should be described in the method how it has been measured so that the VA measurement of the first eye does not influence, improving if it is with the same test, in the second one.

#6 In addition, the inclusion of two eyes of some patients yes, and others no, influences biasing the results in the changes of the aberrations.

#7 What was mentioned in points 5 and 6 should be included as another limitation of the study.

Reviewer 3 Report

I have some comments to the authors:

1. please clearly indicate the retrospective nature and period of analysis in the abstract.

2. the first sub-chapter of the material and methods, "General: " should be a replacement to "Study design."

3. Please add if it was a single- or multi-centered study.

4. please clearly describe the Ethic statement in the material and methods section or why it was not obtained.

5. I think the section "Data Analysis including Vector Analysis: " might change to "Statistical analysis."

6. figures and tables have to be self-explanatory. Please explain What do the lines on the posts mean? SD?

7. please clearly indicate in the text and under the figures/tables which statistical test was to evaluate if p < 0.05.

8. the paper should be corrected and prepared according to the journal's requirements.

Round 2

Reviewer 2 Report

Thank you for the corrections you have made.

The manuscript has been improved according to the recommendations.

Author Response

Thank you very much for your recommendations which have certainly improved our manuscript.

Reviewer 3 Report

I have only minor suggestions, because I am not sure one point.

1. Please clarify if Datagraph-Med is a statistical analysis. In lines 87-89, the authors wrote that they used "Excel", hoverer it should be rather "EXCEL"

to perform statistical analysis.  According to my experience, EXCEL is great for showing primary statistics, not for sophisticated statistical analysis. Please also show the result of the Shapiro-Wilk test. 

It is absolutely mandatory.

2. English should be corrected.

3. Point 4 from the author's reply I wrote this comment depending on my own experience
